# Genetic Gain for Grain Micronutrients and Their Association with Phenology in Historical Wheat Cultivars Released between 1911 and 2016 in Pakistan

**Muzzafar Shaukat** [1], **Mengjing Sun** [2], **Mohsin Ali** [2], **Tariq Mahmood** [1], **Samar Naseer** [1], **Saman Maqbool** [1], **Shoaib Rehman** [3], **Zahid Mahmood** [4], **Yuanfeng Hao** [2], **Xianchun Xia** [2], **Awais Rasheed** [1,2,5,*] and **Zhonghu He** [2,5,*]

1    Department of Plant Sciences, Quaid-i-Azam University, Islamabad 45320, Pakistan; m.muzaffar.shoukat@gmail.com (M.S.); tmahmood@qau.edu.pk (T.M.); samar_naseer@yahoo.com (S.N.); samanmaqbool@bs.qau.edu.pk (S.M.)

2    Institute of Crop Science, Chinese Academy of Agricultural Sciences (CAAS), 12 Zhongguanccun South Street, Beijing 100081, China; smj970626@163.com (M.S.); mali1990@yahoo.com (M.A.); haoyuanfeng@caas.cn (Y.H.); xiaxianchun@caas.cn (X.X.)

3    Institute of Biotechnology, Muhammad Nawaz Shareef University of Agriculture, Multan 66000, Pakistan; shoaib.rehman10@hotmail.com

4    Wheat Program, Institute of Crop Sciences, National Agriculture Research Center (NARC), Park Road, Islamabad 45320, Pakistan; zeearid@gmail.com

5    CIMMYT-China c/o CAAS, 12 Zhongguanccun South Street, Beijing 100081, China

\*    Correspondence: arasheed@qau.edu.pk (A.R.); z.he@cgiar.org (Z.H.)

**Abstract:** Wheat (*Triticum aestivum* L.), being a staple food crop, is an important nutritional source providing protein and minerals. It is important to fortify staple cereals such as wheat with essential minerals to overcome the problems associated with malnutrition. The experiment was designed to evaluate the status of 11 micronutrients including grain iron (GFe) and zinc (GZn) in 62 wheat cultivars released between 1911 and 2016 in Pakistan. Field trials were conducted over two years and GFe and GZn were quantified by both inductively coupled plasma optical emission spectroscopy (ICP-OES) and energy-dispersive X-ray fluorescence spectrophotometer (EDXRF). The GZn ranged from 18.4 to 40.8 mg/kg by ED-XRF and 23.7 to 38.8 mg/kg by ICP-OES. Similarly, GFe ranged from 24.8 to 44.1 mg/kg by ICP-OES and 26.8 to 36.6 mg/kg by EDEXR. The coefficient of correlation was higher for GZn (r = 0.90), compared to GFe (r = 0.68). Modern cultivars such as Zincol-16 and AAS-2011 showed higher GFe and GZn along with improved yield components. Old wheat cultivars WL-711, C-518 and Pothowar-70, released before 1970, also exhibited higher values of GFe and GZn; however, their agronomic performance was poor. Multivariate analysis using eleven micronutrients (Fe, Zn, Al, Ca, Cu, K, Mg, Mn, Na, Se and P) along with agronomic traits, and genome-wide SNP markers identified the potential cultivar with improved yield, biofortification and wider genetic diversity. Genetic gain analysis identified a significant increase in grain yield (0.4% year$^{-1}$), while there was negative gain for GFe ($-0.11\%$ year$^{-1}$) and GZn ($-0.15\%$ year$^{-1}$) over the span of 100 years. The Green Revolution *Rht-B1* and *Rht-D1* genes had a strong association with plant height and grain yield (GY), while semi-dwarfing alleles had a negative effect on GFe and GZn contents. This study provided a valuable insight into the biofortification status of wheat cultivars deployed historically in Pakistan and is a valuable source to initiate a breeding strategy for simultaneous improvement in wheat phenology and biofortification.

**Keywords:** wheat; biofortification; iron; zinc; rht genes

## 1. Introduction

More than two billion people in the world are severely affected by the dietary deficiency of essential micronutrients such as zinc (Zn) and iron (Fe) [1]. Zinc deficiency

leads to stunted growth and increased risk of child mortality, and currently 17% of the global population is at risk of inadequate Zn intake. Fe deficiency leads to anemia, which currently affects 800 million women and children. Other trace elements such as copper (Cu), manganese (Mn), calcium (Ca) and selenium (Se) are also essential micronutrients because they take part in key metabolic reactions for both plant growth and human health [2]. The deficiency of Cu and Mn is not widespread among humans, but these trace elements play a critical role in growth and development. Wheat (*Triticum aestivum* L.) is one of the most important cereal crop plants worldwide and is ranked the third largest producing crop, followed by rice and maize [3]. Wheat provides more than 20% of the calories for the global population, especially for those living in developing countries. Therefore, increasing the micronutrient contents, known as biofortification, in wheat cultivars is a low-cost and sustainable strategy for alleviating micronutrient malnutrition.

Wheat cultivars usually have a low amount of micronutrients, including Fe and Zn [4]. It has been estimated that Zn concentration in wheat grain should be > 50 μg per gram of dry weight, while current wheat grains contain about 25–30 μg Zn per gram dry weight on average [5]. The preliminary breeding target for primary target countries, Pakistan and northern India, is to increase Zn levels by 12 mg/kg, about 50% above the baseline, which is the mean of popular varieties currently grown in the region [4]. Velu et al. [4] concluded that dietary supplements and agronomic practices involving the use of Fe- and Zn-containing fertilizers can help address the nutrient deficiency problem. However, a sustainable and cost-effective approach to increasing essential mineral concentration is through genetic biofortification, which requires identification of cultivars with useful genetic variability for grain minerals and understanding of the physiological and genetic architecture of these minerals in wheat [6].

There is clear evidence that modern and old wheat cultivars differs significantly for grain micronutrients, and it was observed that 23–27% Fe, 33–49% Zn, 25–39% Cu and 29–27% Mg concentrations in grains were decreased after 1965 with the introduction of semi-dwarf and high-yielding wheat cultivars [7]. It is likely due to the fact that *Rht-B1b* and *Rht-D1b* genotypes have reduced growth of the root system, affecting the ability to scavenge minerals from the soil, or the ability to store minerals in the vegetative tissues prior to redistribution to the grain [8]. Murphy et al. [9] analyzed the mineral elements in 63 historical wheat cultivars released between 1842 and 1965 in Pacific Northwest US and concluded that all minerals except Ca significantly decreased over time. Although breeding for biofortified wheat is not a main target in Pakistan and India, high-Zn cultivars such as Zincol-2016 and Zn Shakti have been released in Pakistan and India, respectively [10]. Previously, some cultivars with high grain Fe/Zn have been released, such as cv. Burnside in Canada with the *Gpc-B1* gene [11]. Therefore, it is very important to analyze the status of minerals in historical wheat germplasm for better insight when selecting germplasm resources in breeding.

The present study was designed to evaluate the status of minerals contents in historical wheat cultivars released in Pakistan between 1911 to 2016. The main objectives included were: (a) to assess the temporal variation in grain mineral elements in historical wheat cultivars and rate of progress for improvement in grain mineral elements, (b) to identify the important phenological traits associated with grain mineral elements in historical wheat cultivars of Pakistan and (c) to identify the allelic effects of important Green Revolution *Rht-1* genes, and others such as *NAM-A1, TaSus2-2B, TaGW2-6B* and *TaGW2-6A*, on agronomic traits and mineral contents in historical wheat cultivars from Pakistan

## 2. Materials and Methods

### 2.1. Plant Material and Field Trials

A set of 62 wheat cultivars released in Pakistan from 1911 to 2016 were selected for this study. The cultivar name, year of release and pedigree are given in Table 1. The cultivars were evaluated for two years, 2018–2019 (later as 2018) and 2019–2020 (later as 2019), in the field at the National Agriculture Research Center (NARC), Islamabad, Pakistan, using

a randomized complete block design (RCBD) with two replications. The NARC site is located at 33°43′ N 73°04′ E. The date of sowing was 5 December in 2018 and 8 December in 2019.

**Table 1.** Pedigree of historical wheat cultivars evaluated for phenological parameters along with the grain iron and zinc content.

| Cultivar | Release Year | Pedigree |
|---|---|---|
| T9 | 1911 | Nil |
| C-518 | 1933 | T9/8A |
| C-217 | 1944 | C516/C591 |
| C-271 | 1957 | C230/IP165 |
| C-273 | 1957 | C209/C591 |
| Dirk | 1958 | FORD//DUNDEE/BOBIN or FORD/DONDEE (I) |
| Mexipak-65 | 1965 | PJ/GB55 or PJ62/GB55 |
| Pothowar-70 | 1970 | BURT/KENYA//QUETA(L)/3/NAD63 |
| Pari-73 | 1973 | CNO67//SN64/KLRE/3/8156 |
| Parula | 1973 | Nil |
| WL-711 | 1978 | S308/CHRIS//KAL |
| Pak-81 | 1981 | KVZ/BUHO//KAL/BB |
| Barani-83 | 1983 | BB/GLL/3/GTO/7C//BB/CNO67 |
| Chakwal-86 | 1986 | FORLANI/ACC//ANA or Fln/ACS//ANA |
| Khyber-87 | 1987 | KVZ/TRM//PTM/ANA |
| Rawal-87 | 1987 | MAYA/MON//KVZ/TRM |
| Inquilab-91 | 1991 | WL 711/CROW "S" |
| Pasban-90 | 1991 | INIA F66/TH.DISTICHUM//INIAF66/3/GENARO T81 |
| Pastor | 1993 | PFAU/SERI-82//BOBWHITE |
| Bakhtawar-94 | 1994 | AU/UP301//GLL/SX/3/PEW/4/MAI/MAYA//PEW |
| Parwaz-94 | 1995 | V.5648/PARULA |
| Punjab-96 | 1996 | SA42*2/4/CC/INIA//BB/3/INIA/HD832 |
| Suleman-96 | 1996 | F6.74/BUN//SIS/3/VEE#7 or F6-74/BUN//SIS/3/VEE#7 |
| Tatara | 1996 | JUP/ALD'S'//KLT'S' |
| Chakwal-97 | 1997 | BUC'S'/FCT'S' |
| MH-97 | 1997 | NORD-DESPREZ(ND)/VG-9144//KALYANSONA/BLUEBIRD/3/YACO/4/VEERY-5 |
| Margallah-99 | 1999 | OPATA/BOW'S' |
| Auqab-2000 | 2000 | CROW'S'/NAC//BOW'S' |
| Wafaq-2001 | 2001 | OPATA/RAYON//KAUZ |
| AS-2002 | 2002 | KHP/D31708//CM74A370/3/CNO79/4/RL6043/4*NAC |
| Bhakkar-2002 | 2002 | P20102/PIMA/SKA/3/TTR'S'/BOW'S' |
| GA-2002 | 2002 | DWL5023/SNB//SNB |
| Ufaq | 2002 | V.84133/V83150 |
| Pirsabak-2004 | 2004 | KAUZ/STAR |
| Pirsabak-2005 | 2005 | MUNIA/CHTO//AMSEL |
| Fareed-2006 | 2006 | PT'S'/3/TOB/LFN//BB/4/BB/HD-832-5//ON/5/G-V/ALD'S'//HPO |
| Seher-2006 | 2006 | CHILL/2*STAR/4/BOW//BUC/PVN/3/2*VEE#10 |
| Bathoor | 2008 | URES/JUN//KAUZ |
| Chakwal-50 | 2008 | ATTILA/3/HUI/CARC//CHEN/CHTO/4/ATTILA |
| Faisalabad-2008 | 2008 | PBW65/2*Pastor |
| Mairaj-2008 | 2008 | SPARROW/INIA//V.7394/WL711/13/BAUS |
| Pirsabak-2008 | 2008 | KAUZ/PASTOR |
| NARC-2009 | 2009 | INQALAB 91*2/TUKURU |
| Atta-Habib | 2010 | INQALAB 91*2/TUKURU |
| Barsat-2009 | 2010 | FRET2 |
| AAS-2011 | 2011 | PRL/PASTOR//2236(V6550/SUTLEH-86) |
| Dharabi-2011 | 2011 | HXL-7573/2*BAGULA//PASTOR |
| Millat-2011 | 2011 | CHENAB2000/INQ-91 |
| NARC-2011 | 2011 | OASIS/SKAUZ//4*BCN/3/2*PASTOR |
| Punjab-2011 | 2011 | AMSEL/ATTILA//INQ-91/PEW'S' |
| Galaxy-2013 | 2013 | PUNJAB-96/V-87094//MH-97 |
| Pakistan-2013 | 2013 | MEX94.27.1.20/3/Sokoll//Attila/3*BCN |
| Pirsabak-2013 | 2013 | CS/TH.SC//3*PVN/3/MIRLO/BUC/4/MILAN/5/TILHI |
| Shahkar-2013 | 2013 | CMH84.3379/CMH78.578//MILAN |
| Pakhtunkhwa-2015 | 2015 | WBLL1*2/4/YACO/PBW65/3/KAUZ*2/TRAP//KAUZ |
| Ujala-2016 | 2015 | KIRITATI/4/2*WEAVER/TSC//WEAVER/3/WEAVER |
| Ahsan-2016 | 2016 | Pastor/3/Altar 84/Ae. squarrosa//Opata |
| Borlaug-2016 | 2016 | Sokoll/3/Pastor//HXL7573/2*BAU |
| Gold-2016 | 2016 | Nil |
| Johar-2016 | 2016 | KAUZ/PASTOR//V.3009 |
| Zincol-2016 | 2016 | OASIS/ SKAUZ//4*BCN/3/2*PASTOR/4/T.SPELTA PI348449/5/BACEU#1/6/ WBLL1*2/CHAPIO |
| Local-White | - | Nil |

## 2.2. Phenotyping

The data were recorded for ten morphological parameters, including tillers per plant (TPP), plant height (PH) in cm, spike length (SL) in cm, spikelet per spike (SNPS), grains per spike (GPS), thousand kernel weight (TKW) in grams, grain length (GL) in mm, grain width (GW) in mm, grain diameter (GD) in mm and grain yield (GY) in t/ha, and were recorded at different growth stages. Plant height from the ground to the top of the spike was recorded at late grain filling (Z77). Number of grains per spike was the mean of randomly selected spikes from 15 different plants per plot. Number of spikes per plot was counted at physiological maturity (Z96), all plants in each plot were harvested manually and above-ground total biomass weight was recorded. Grain yield was measured as weight of grain harvested per plot. Thousand kernel weight was based on three 200-grain samples from each plot. GL and GW were recorded for one hundred uniform seeds from each plot. Eleven mineral concentrations in the grains of sixty-two wheat cultivars were determined from each year and each replicate using inductively coupled plasma mass

spectroscopy (ICP-OES). Briefly, grain samples were digested with concentrated $HNO_3$, hydrogen peroxide and hydrofluoric acid. The grain Fe and Zn were further evaluated with non-destructive high-throughput ED-XRF (energy-dispersive X-Ray fluorescence analysis) method. This was performed by using an Oxford Instruments X-Supreme 8000 fitted with a ten place auto-sampler holding 40 mm aluminum cups. Fe and Zn were quantified and analyzed in 186 s in which the acquisition time was 60 s and dead time was also 60 s [12]. All the analyses were done at the central facility of the Institute of Crop Science, Chinese Academy of Agricultural Sciences (CAAS), Beijing, China.

### 2.3. Genotyping Using KASP Markers and GBTS

Total genomic DNA was extracted from each cultivar following a previously described protocol [13]. The KASP markers for genes *Rht-B1, Rht-D1, TaSus2-2B, TaGW2-6A,* and *TaGW2-6B* were used from our previous work [14]. Two KASP markers for NAM-A1 were used from Cormier et al. [15]. The PCR mix included 2 μL of 50–100 ng/μL template DNA, 2.5 μL of 2X KASP master mix, 0.07 μL of KASP assay mix and 2.5 μL of distilled water. PCR was performed in 384-well formats (S1000, Thermal Cycler, USA) by the following procedure: hot start at 95 °C for 15 min, followed by 10 touchdown cycles (95 °C for 20 s; touchdown at 65 °C initially and decreasing at $-1$ °C per cycle for 25 s) and then 30 additional cycles of annealing (95 °C for 10 s; 57 °C for 60 s).

DNA samples were also genotyped with a genotyping-by-targeted-sequencing (GBTS) platform which included more than 100 SNPs distributed over all the wheat chromosomes. This assay was used for genetic diversity studies among cultivars. The design for the GBTS assay is not reported yet.

### 2.4. Statistical Analysis

For each trait, the best linear unbiased estimator (BLUE) for each genotype was estimated using a mixed linear model across two environments. The full model was as follows:

$$Y_{ijkl} = \mu + geno_i + env_j + geno \times env_{ij} + Rep(env)_{jk} + \varepsilon_{ijkl}$$

where $Y_{ijkl}$ is the average phenotype of an individual plot, μ is the grand mean, $geno_i$ is the fixed effect of the *i*th genotype, $env_j$ is the random effect of the *j*th env (year in this case), $geno \times env_{ij}$ is the random effect of interaction between the *i*th genotype and the *j*th env, $Rep(env)_{jk}$ is the random effect of the *k*th Rep nested within the *j*th env and $\varepsilon_{ijkl}$ is the residual effect that was assumed to be independent and identically distributed following a normal distribution with a mean of zero and variance $\sigma_\varepsilon^2$. After removing the outliers for each phenotype, an iterative mixed linear model was fitted in *lmerTest* (R package) with the full model. For each phenotype, the model was used to calculate the BLUE values for each genotype and estimate the variance components for broad-sense heritability ($H^2$) on a line basis, as well as standard error by delta method.

To assess the degree of the association between BLUE values for each pair of traits, the pairwise Pearson's correlation coefficients were estimated via R package *Hmisc* and plotted using R package *PerforanceAnalytics*. The mean, range and standard deviation of BLUEs (Table S2) were calculated via R statistics standard functions in the *stats* and the *pysch* packages.

For each trait, the genetic gain over time was estimated by a simple linear regression in R function *lm*. The model was as follows:

$$y_i = \beta_0 + \beta_1 x_i + \varepsilon_i$$

where $y_i$ is the BLUE value of the *i*th genotype, xi is the year of release of the *i*th genotype, $\beta_0$ is the intercept of the regression line, $\beta_1$ is the regression coefficient and $\varepsilon_i$ is the residual effect. The regression coefficient was used to estimate the genetic gain [16]. The *t*-test on the regression coefficient $\beta_1$ was carried out to examine the significance of the regression with the null hypothesis: $\beta_1 = 0$, and the significance level was 0.05.

SNP markers from GBTS assay were used to conduct principal component analysis (PCA) and estimates of unweighted paired group arithmetic mean (UPGMA) based on the neighbor-joining (NJ) method using TASSEL version 5.0. The SNP markers with more than 5% missing data and minor allele frequency were removed. The KASP markers were used to assess the allelic effects on the individual traits using Student's *t*-test.

## 3. Results

The historical wheat cultivars showed significant variations in morphological traits and the 11 micronutrients evaluated in this study. Moreover, the allelic effects of some of the genes were also significant on the micronutrient traits. The results are given below in each subsection.

### 3.1. Variation in Micronutrients and Morphological Traits

All 62 wheat cultivars were grouped into three categories according to the year of release. The first group included seven cultivars released during 1911 to 1965, the second group included 21 cultivars released during 1965–2000 and the third group included 34 cultivars released after 2000. Descriptive statistics for all morphological traits and micronutrients across three groups are described in Table 2. The coefficient of variation was highest for Na contents (75.4%), and the lowest CV% was observed for GD (3.7%). The mean and range for all traits in cultivars released in three breeding eras are also described in Table 2. The GY progressively increased from 1.3 to 2.09 and 2.44 t/ha in three breeding eras, respectively. Contrastingly, TKW was higher (43.4 g) in old cultivars, compared to 39.1 g mid-era and 41.3 g in cultivars released after 2000 (Table 2; Figure 1). Similarly, PH was 111 cm in old cultivars, 97.6 cm in cultivars released in 1965–2000 and 93.7 cm in post-2000 cultivars. Among the micronutrients, there was no clear pattern for GFe and GZn; however, K was significantly higher and Se, Mg and Cu were significantly higher in cultivars released before 1965 (Table 2; Figure 1).

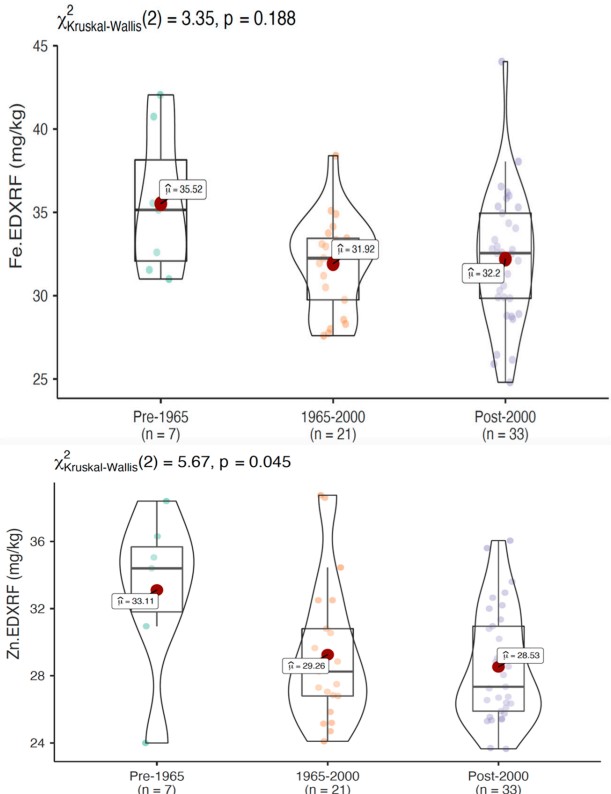

**Figure 1.** *Cont.*

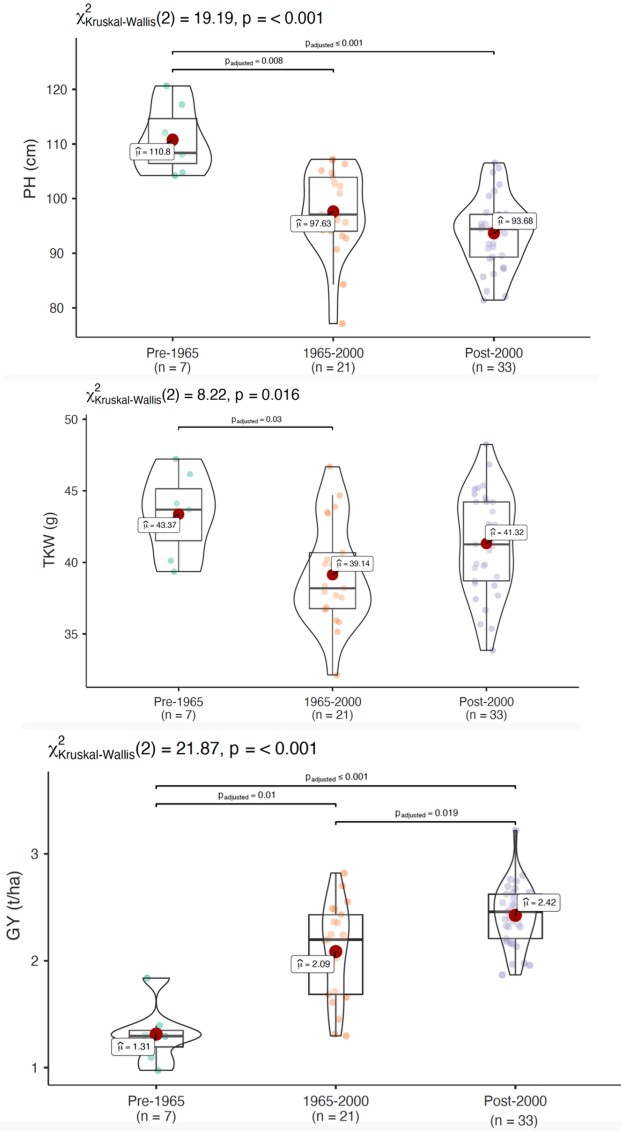

**Figure 1.** Box plots showing the variation among important micronutrients and yield-related traits distributed over three selective breeding periods. The significance between mean performance of three breeding periods is shown as *p* value of Kruskal–Wallis test.

Analysis of variance (ANOVA) showed variation in nearly all the traits except grain Se contents (Table 3). All the traits showed significant variations by genotype, year and genotype x year interaction, with some exceptions. No significant variation was observed by planting year in TPP, PH, TKW, GL, GW, Al, Ca, Cu and Se content (Table 3). Among the micronutrients, the heritability for Fe and Zn was 0.62 and 0.68, respectively (Table 3). The heritability for important yield-related traits such as TKW, TPP and GY was 0.66, 0.35 and 0.68, respectively.

**Table 2.** Descriptive statistics of grain mineral elements and morphological traits in sixty-two wheat cultivars classified into three distinct breeding eras.

| | Pre-1965 (*n* = 7) | | | | 1965–2000 (*n* = 21) | | | | Post-2000 (*n* = 34) | | | | Overall (*n* = 62) | | | |
|---|---|---|---|---|---|---|---|---|---|---|---|---|---|---|---|---|
| Traits | Min | Mean | Mean | CV(%) | Min | Max | Mean | CV(%) | Min | Max | Mean | CV(%) | Min | Max | Mean | CV (%) |
| Fe.EDXRF (mg/kg) | 31 | 42 | 35.5 | 12.34 | 27.6 | 38.4 | 31.9 | 8.9 | 24.8 | 44 | 32.2 | 12.39 | 24.8 | 44 | 32.5 | 11.66 |
| Zn.EDXRF (mg/kg) | 24 | 38.4 | 33.1 | 14.14 | 24.1 | 38.8 | 29.3 | 14.2 | 23.6 | 36 | 28.5 | 11.72 | 23.6 | 38.8 | 29.3 | 13.62 |
| Fe.ICPOES (mg/kg) | 30.5 | 38.6 | 32.7 | 8.96 | 26.9 | 37.3 | 32.8 | 6.83 | 26.8 | 37.6 | 33.7 | 6.5 | 26.8 | 38.6 | 33.3 | 6.91 |
| Zn.ICPOES (mg/kg) | 18.8 | 33.1 | 28.8 | 16.63 | 20.4 | 40.8 | 29.8 | 14.7 | 23.9 | 37.2 | 28.5 | 11.82 | 18.8 | 40.8 | 28.9 | 13.43 |
| Al (mg/kg) | 2.69 | 4.78 | 3.87 | 21.06 | 2.44 | 7.78 | 4.48 | 23.21 | 2.06 | 4.97 | 3.58 | 18.91 | 2.06 | 7.78 | 3.92 | 23.37 |
| Ca (mg/kg) | 453 | 635 | 547 | 11.99 | 476 | 701 | 588 | 9.73 | 415 | 720 | 563 | 11.26 | 415 | 720 | 570 | 10.91 |
| Cu (mg/kg) | 3.81 | 5.28 | 4.68 | 10.51 | 3.25 | 5.69 | 4.16 | 17 | 3.06 | 5.47 | 4.08 | 14.24 | 3.06 | 5.69 | 4.18 | 15.22 |
| K (mg/kg) | 3591 | 4118 | 3892 | 4.83 | 3762 | 4891 | 4385 | 6.07 | 3732 | 5344 | 4279 | 8.72 | 3591 | 5344 | 4271 | 8.19 |
| Mg (mg/kg) | 984 | 1288 | 1181 | 8.64 | 923 | 1318 | 1134 | 8.06 | 970 | 1293 | 1130 | 7.22 | 923 | 1318 | 1137 | 7.69 |
| Mn (mg/kg) | 24.4 | 37.4 | 32.2 | 13.45 | 25.6 | 37.8 | 32 | 9.53 | 26.3 | 38.2 | 32.7 | 9.88 | 24.4 | 38.2 | 32.4 | 10.06 |
| Na (mg/kg) | 8.75 | 47.3 | 28.7 | 55.4 | 10 | 159 | 43.1 | 87.01 | 8.94 | 79.9 | 32.1 | 59.5 | 8.75 | 159 | 35.4 | 75.42 |
| P (mg/kg) | 2752 | 3349 | 3108 | 7.53 | 2862 | 3918 | 3361 | 7.94 | 2719 | 3657 | 3150 | 7.71 | 2719 | 3918 | 3217 | 8.33 |
| Se (mg/kg) | 0.199 | 0.258 | 0.232 | 8.84 | 0.132 | 0.24 | 0.199 | 18.69 | 0.135 | 0.26 | 0.212 | 14.29 | 0.132 | 0.26 | 0.21 | 15.71 |
| TPP | 3.13 | 4.25 | 3.58 | 12.04 | 2.83 | 4.34 | 3.53 | 12.32 | 2.44 | 4.08 | 3.23 | 11.8 | 2.44 | 4.34 | 3.37 | 12.76 |
| PH (cm) | 104 | 121 | 111 | 5.6 | 77.1 | 107 | 97.6 | 9.54 | 81.4 | 107 | 93.7 | 11.11 | 77.1 | 121 | 97 | 8.85 |
| SL | 12.8 | 18.5 | 15.8 | 12.72 | 15.1 | 22.2 | 17.2 | 9.42 | 14.1 | 19.2 | 16.9 | 6.98 | 12.8 | 22.2 | 16.9 | 8.70 |
| SNPS | 18.1 | 20.3 | 19.3 | 4.13 | 18.1 | 20.9 | 19.7 | 3.88 | 17.6 | 21.1 | 19.2 | 4.29 | 17.6 | 21.1 | 19.4 | 4.27 |
| GPS | 39.3 | 52.3 | 45.9 | 11.18 | 48.6 | 61.3 | 53.3 | 6.42 | 44.1 | 64 | 54.2 | 8.03 | 39.3 | 64 | 53 | 9.11 |
| GY (t/ha) | 0.975 | 1.84 | 1.31 | 20.76 | 1.3 | 2.82 | 2.09 | 21.77 | 1.87 | 3.22 | 2.44 | 12.05 | 0.975 | 3.22 | 2.19 | 22.74 |
| TKW (g) | 39.4 | 47.2 | 43.4 | 6.66 | 32.1 | 46.7 | 39.1 | 9.26 | 33.9 | 48.2 | 41.3 | 8.45 | 32.1 | 48.2 | 40.8 | 9.02 |
| GL (mm) | 1.11 | 1.36 | 1.25 | 7.01 | 1.16 | 2.04 | 1.46 | 14.11 | 1.16 | 2.23 | 1.38 | 16.09 | 1.11 | 2.23 | 1.39 | 15.32 |
| GW (mm) | 1.97 | 2.44 | 2.11 | 7.91 | 2.15 | 2.73 | 2.35 | 5.62 | 2.02 | 2.52 | 2.27 | 5.55 | 1.97 | 2.73 | 2.27 | 6.56 |
| GD (mm) | 5.86 | 6.65 | 6.16 | 4.69 | 5.89 | 6.65 | 6.36 | 3.25 | 5.77 | 6.8 | 6.38 | 3.65 | 5.77 | 6.8 | 6.34 | 3.77 |

Fe calculated by EDXRF, Zn calculated by EDXRF, Fe calculated by ICPOES and Zn calculated by ICPOES. Aluminum (Al), calcium (Ca), copper (Cu), potassium (K), magnesium (Mg), manganese (Mn), sodium (Na), phosphorous (P) and selenium (Se). Plant height (PH), grains per spike (GPS), thousand kernel weight (TKW), tillers per plant (TPP), spike length (SL), spike number per spike (SNPS), grain number per spike (GpS), grain yield (GY), grain length (GL), grain width (GW), grain diameter (GD).

**Table 3.** Analysis of variance for morphological traits and micronutrient contents in historical wheat cultivars of Pakistan.

| | Replication | Genotype (G) | Year (Y) | G x Y Interaction | |
| --- | --- | --- | --- | --- | --- |
| **df** | **2** | **61** | **1** | **61** | |
| **Traits** | | **Means Squares** | | | **Heritability** |
| TPP | 1.162 * | 0.7386 *** | 0.0413 ns | 0.9095 *** | 0.35 |
| PH | 203.1443 ns | 465.0146 *** | 256.8787 ns | 156.766 | 0.67 |
| SL | 1.2842 ns | 8.6695 *** | 13.518 * | 5.5964 ** | 0.49 |
| SNPS | 222.2401 *** | 4.5472 *** | 141.0344 *** | 1.6726 ns | 0.48 |
| GNPS | 675.4824 *** | 93.4753 *** | 598.6102 *** | 51.7566 ns | 0.52 |
| GY | 0.1032 *** | 0.9954 *** | 2.3278 *** | 0.1575 ** | 0.68 |
| TKW | 0.3556 ns | 54.2871 *** | 0.0429 ns | 19.7962 ns | 0.66 |
| GL | 7.9117 *** | 0.2239 *** | 0.0032 ns | 0.0451 ns | 0.80 |
| GW | 8.6088 *** | 0.1296 *** | 0.0147 ns | 0.0455 ns | 0.65 |
| GD | 8.329 *** | 0.2752 *** | 0.0022 ns | 0.0579 ns | 0.78 |
| Al | 68.6896 *** | 3.1559 *** | 0.127 ns | 0.3507 ns | 0.52 |
| Ca | 35,341.2148 *** | 11,898.9395 *** | 81.7116 ns | 75.8745 ns | 0.86 |
| Cu | 8.2654 *** | 1.1161 *** | 0.5172 ns | 0.066 ns | 0.72 |
| Fe.EDXRF | 4.5506 ** | 40.2081 *** | 0.1301 ns | 37.163 *** | 0.51 |
| Fe.ICPOES | 14.5627 *** | 15.3527 *** | 120.9392 *** | 8.7979 *** | 0.62 |
| K | 112,509.1797 * | 455,115.7812 *** | 756,099.6875 *** | 113,190.0078 *** | 0.79 |
| Mg | 6691.2295 * | 25,110.6953 *** | 52,088.6211 *** | 8330.1592 *** | 0.73 |
| Mn | 5.0568 * | 35.1404 *** | 5.8521 * | 11.7127 * | 0.74 |
| Na | 23.8786 ns | 2031.3976 *** | 6190.4634 *** | 2507.9097 *** | 0.43 |
| P | 49,623.0234 * | 205,982.9375 *** | 390,194.75 *** | 120,418.9297 *** | 0.61 |
| Zn.EDXRF | 11.2164 ** | 39.5774 *** | 11.9913 * | 34.2843 *** | 0.52 |
| Zn.ICPOES | 8.4438 ** | 41.8175 *** | 594.0781 *** | 18.7137 *** | 0.68 |
| Se | 7.3049 *** | 0.045 ns | 0.0319 ns | 0.0518 ns | 0.0057 |

\*, significant ($p < 0.05$); \*\*, significant ($p < 0.01$); \*\*\*, significant ($p < 0.001$); ns, non-significant ($p > 0.05$).

Among the micronutrients, two methods were used to phenotype GFe and GZn. GFe ranged between 24.8 (Auqab-2000) and 44.0 mg/kg (Zincol-16), with an average of 32.5 mg/kg using EDXRF, while it ranged from 26.8 to 38.6 mg/kg with an average of 33.3 mg/kg with ICP-OES (Table 2). The correlation coefficient between the two methods was r = 0.53 (Figure 2). Similarly, the GZn content ranged between 23.65 (Auqab-2000) and 38.8 mg/kg (WL-711) with a mean value of 29.30 mg/kg by EDXRF, and ranged from 18.4 (Rawal-87) to 40.8 mg/kg (Pothowar-70) with an average of 28.9 mg/kg by ICP-OES. The correlation coefficient between the two methods was r = 0.82 (Figure 2). Other important micronutrients such as Ca ranged from 415 to 720 mg/kg with an average of 570 mg/kg. Similarly, Mn ranged between 24.4 and 38.2 mg/kg with an average of 32.4 mg/kg (Table 2).

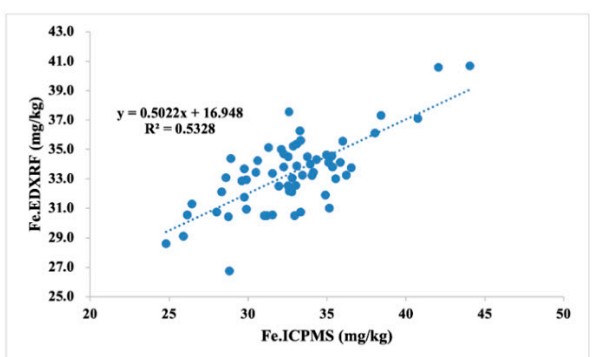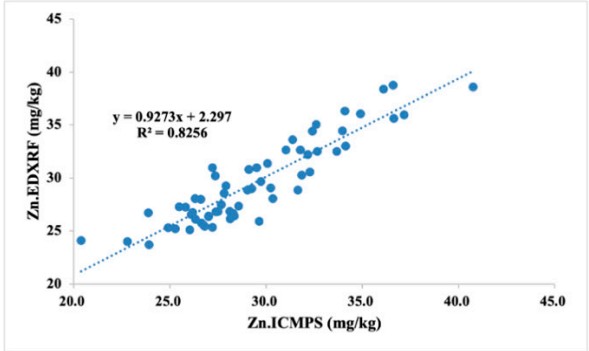

**Figure 2.** Coefficient of determination between two ICP-OES and EDXRF methods to predict grain iron (GFe) and zinc (Zn) in historical wheat cultivars released between 1911 and 2016.

### 3.2. Correlation between Traits and Multivariate Analysis

The coefficient of correlation is reported in Table 4 between all morphological traits and micronutrients. The coefficient of correlation between GFe and GZn was positive with r = 0.31. GFe had a strong positive correlation with Ca (r= 0.28), Mg (r = 0.35), Mn (r = 0.3) and Se (r = 0.26), while its correlation was non-significant with any morphological trait. GZn had a relatively higher correlation with Ca (r = 0.61), Cu (r = 0.6), K (r = 0.4), Mg (r = 0.74) and P (r = 0.63), while GZn had a strong negative correlation with GD (r = −0.48). GY had a negative correlation with TPP (r = −0.3) and a positive correlation with GPS (r = 0.4). The highest correlation among the morphological traits was between SL and SNPS (r = 0.46).

The PCA biplot clearly separated the cultivars into three groups consistent with the three breeding eras defined previously (Figure 3a). The first two principal components explained 19 and 12.1% of the total variation. The pre-1965 cultivars were separated on the lower side of the PC2 in admixture, containing cultivars Chakwal-86 and Rawal-87. The dendrogram showed two major clusters, cluster I and cluster II. Cluster I consisted of 20 cultivars mostly released after 1965, except Dirk, while cluster II consisted of 42 cultivars and was further subdivided into three subclusters. The clustering was consistent with the breeding eras except that C-273 was in admixture with some modern cultivars (Figure 3b). The dendrogram generated from the genome-wide SNP marker also corroborated the diversity pattern showed by the phenotypic analysis (Figure 3c).

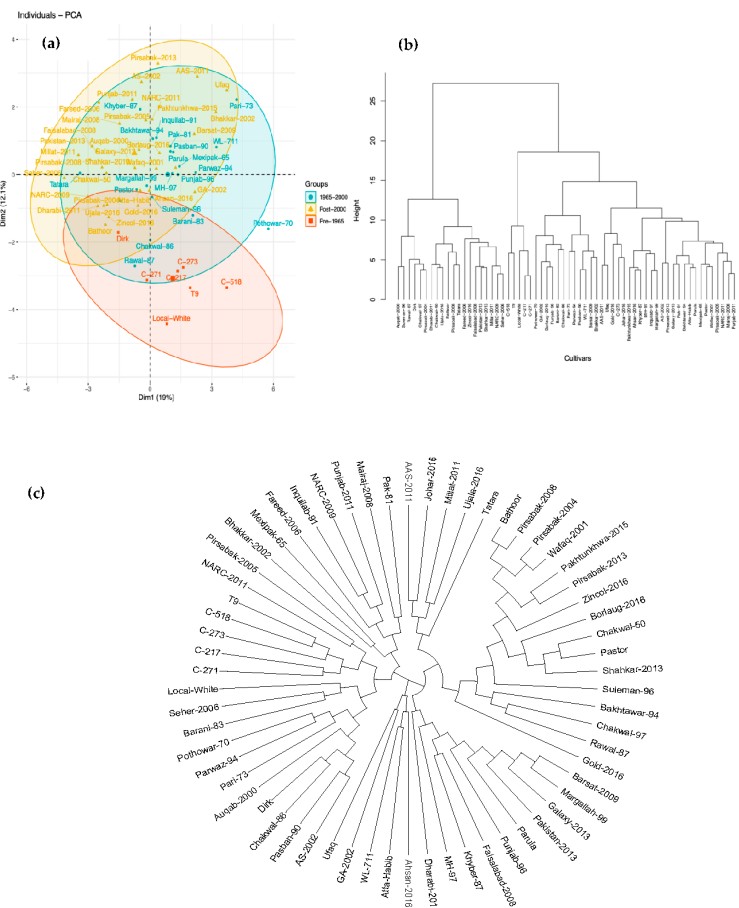

**Figure 3.** Multivariate analysis of historical wheat cultivars based on morphological traits and micronutrients, and genome-wide SNP markers. (**a**) Principal components analysis-based biplot showing scattering of wheat cultivars grouped based on three breeding periods, (**b**) dendrogram showing clustering of wheat cultivars based on morphological traits and micronutrients, (**c**) dendrogram showing clustering of wheat cultivars based on genome-wide SNP markers.

**Table 4.** Coefficient of correlation between morphological traits and grain micronutrients in historical wheat cultivars of Pakistan.

| Traits | Fe.ICPOES (mg/kg) | Zn.ICPOES (mg/kg) | Al (mg/kg) | Ca (mg/kg) | Cu (mg/kg) | K (mg/kg) | Mg (mg/kg) | Mn (mg/kg) | Na (mg/kg) | P (mg/kg) | Se (mg/kg) | TPP | PH (cm) | SL (cm) | SNPS | GPS | GY (t/ha) | TKW (g) | GL (cm) | GW (cm) |
|---|---|---|---|---|---|---|---|---|---|---|---|---|---|---|---|---|---|---|---|---|
| Zn.ICPOES (mg/kg) | 0.313 * | — | | | | | | | | | | | | | | | | | | |
| Al (mg/kg) | −0.041 | 0.166 | — | | | | | | | | | | | | | | | | | |
| Ca (mg/kg) | 0.278 * | 0.613 *** | 0.234 | — | | | | | | | | | | | | | | | | |
| Cu (mg/kg) | 0.197 | 0.601 *** | 0.359 ** | 0.378 ** | — | | | | | | | | | | | | | | | |
| K (mg/kg) | 0.149 | 0.399 ** | 0.13 | 0.551 *** | 0.188 | — | | | | | | | | | | | | | | |
| Mg (mg/kg) | 0.354 ** | 0.747 *** | 0.058 | 0.521 *** | 0.536 *** | 0.278 * | — | | | | | | | | | | | | | |
| Mn (mg/kg) | 0.299 * | 0.24 | −0.121 | 0.226 | 0.088 | 0.042 | 0.412 *** | — | | | | | | | | | | | | |
| Na (mg/kg) | −0.01 | 0.299 * | 0.286 * | 0.253 * | 0.331 ** | 0.38 ** | 0.321 * | −0.171 | — | | | | | | | | | | | |
| P (mg/kg) | 0.126 | 0.637 *** | 0.195 | 0.498 *** | 0.417 *** | 0.555 *** | 0.667 *** | 0.353 ** | 0.279 * | — | | | | | | | | | | |
| Se (mg/kg) | 0.26 * | 0.373 ** | 0.132 | 0.402 ** | 0.42 *** | 0.174 | 0.536 *** | 0.296 * | 0.302 * | 0.101 | — | | | | | | | | | |
| TPP | −0.134 | 0.11 | 0.355 ** | −0.083 | 0.315 * | 0.027 | 0.043 | −0.071 | 0.281 * | 0.203 | −0.024 | — | | | | | | | | |
| PH (cm) | −0.093 | 0.114 | −0.004 | −0.094 | 0.291 * | −0.317 * | 0.114 | 0.053 | 0.051 | 0.004 | 0.131 | 0.296 * | — | | | | | | | |
| SL (cm) | 0.156 | 0.191 | −0.019 | 0.187 | −0.018 | 0.118 | 0.09 | 0.084 | −0.02 | 0.282 * | −0.127 | 0.181 | 0.252 * | — | | | | | | |
| SNPS | −0.041 | 0.153 | 0.207 | 0.078 | 0.114 | 0.019 | −0.07 | −0.055 | 0.059 | 0.148 | −0.059 | 0.359 ** | 0.344 ** | 0.459 *** | — | | | | | |
| GPS | 0.113 | 0.012 | −0.046 | 0.02 | −0.065 | 0.211 | −0.137 | 0.103 | −0.069 | 0.123 | −0.147 | −0.111 | −0.09 | 0.282 * | 0.457 *** | — | | | | |
| GY (t/ha) | 0.2 | −0.149 | −0.354 ** | −0.176 | −0.462 *** | 0.07 | −0.227 | 0.17 | −0.169 | −0.108 | −0.248 | −0.329 ** | −0.216 | 0.187 | −0.073 | 0.393 ** | — | | | |
| TKW (g) | −0.005 | −0.084 | −0.152 | −0.103 | −0.013 | −0.093 | 0.124 | 0.111 | −0.042 | −0.117 | 0.324 * | −0.157 | 0.175 | −0.015 | −0.257 * | −0.115 | 0.011 | — | | |
| GL (cm) | 0.205 | 0.072 | 0.053 | 0.141 | −0.166 | 0.108 | 0.113 | −0.145 | −0.03 | 0.162 | −0.07 | −0.108 | −0.027 | 0.195 | −0.023 | 0.061 | 0.166 | −0.128 | — | |
| GW (cm) | 0.19 | 0.129 | 0.211 | 0.331 * | 0.17 | 0.503 *** | 0.067 | −0.238 | 0.301 * | 0.229 | 0.11 | −0.026 | −0.185 | 0.173 | 0.077 | 0.232 | 0.086 | −0.124 | 0.37 ** | — |
| GD (cm) | −0.059 | −0.488 *** | −0.081 | −0.372 *** | −0.404 ** | −0.027 | −0.309 | −0.044 | −0.004 | −0.179 | −0.246 | 0.045 | 0.026 | 0.179 | −0.073 | 0.259 * | 0.359 ** | 0.304 * | 0.215 | 0.278 * |

*, significant ($p < 0.05$); **, significant ($p < 0.01$); ***, significant ($p < 0.001$); ns, non-significant ($p > 0.05$).

### 3.3. Genetic Gain for Micronutrients and Morphological Traits

The genetic gain analysis identified the traits that significantly changed with the release year (Table S1). Among the morphological traits, there was significant change in TPP, PH and SNPS, which reduced significantly over time, while GY and GpS significantly increased over time. The highest yielding cultivar, Punjab-2011 (3.2 t/ha), yielded almost thrice that of the lowest yielding cultivar T9 (0.97 t/ha). The increase in genetic gain was 0.41% over the period of 105 years, while the increase was highest in the recent period after 2000. The change in TKW, GL and GW remained non-significant over the years. Among the micronutrients, GZn significantly reduced during the course of breeding to $-0.05$ mg/kg/year (0.12%), while GFe also reduced at the rate of $-0.02$ mg/kg/year, but the change was non-significant.

### 3.4. Allelic Variation in Functional Genes and Association with Traits

The KASP markers for six genes were used to identify the allelic variation in historical wheat cultivars. The *Rht-B1* and *Rht-D1* genes were combined to identify the *Rht-1* haplotypes in the wheat cultivars. The results reveal that semi-dwarfing alleles, either *Rht-B1b* or *Rht-D1b*, were introduced after 1965 and their frequency was 79%, compared to the 21% frequency of the *Rht-B1a/Rht-D1a* haplotype (Table 5). Wheat sucrose synthase gene, *TaSus2-2B*, had two haplotypes and haplotype Hap-L had a very high frequency of 85.5% compared to 14.5% of the Hap-H frequency. Similarly, two grain-width-related genes, *TaGW2-6A* and 6B, were also surveyed. The frequency of haplotypes associated with higher TKW of Hap-I was 32% at *TaGW2-6B*, while the frequency of Hap-6A-A was 80.6%, which was associated with higher TKW. At the *NAM-A1* locus, two haplotypes *NAM-A1b* and *NAM-A1d* had frequencies of 40.3 and 59.7%, respectively.

**Table 5.** Allelic frequencies in percentages for important functional genes in historical wheat cultivars of Pakistan.

| Genes | Alleles | Cultivar Release Era | | | |
| --- | --- | --- | --- | --- | --- |
| | | **Pre-1965** | **1965–2000** | **Post-2000** | **Overall** |
| *Rht-8* | *Rht8* | 8.1 | 30.6 | 50.0 | 88.7 |
| | *rht-8* | 3.2 | 3.2 | 4.8 | 11.3 |
| *Rht-1* | *Rht-B1a/Rht-D1a* | 9.7 | 6.5 | 4.8 | 21.0 |
| | *Rht-B1a/Rht-D1b* | 0.0 | 4.8 | 8.1 | 12.9 |
| | *Rht-B1b/Rht-D1a* | 0.0 | 24.2 | 41.9 | 66.1 |
| *TaSus2-2B* | Hap-H | 11.3 | 1.6 | 1.6 | 14.5 |
| | Hap-L | 0.0 | 32.3 | 53.2 | 85.5 |
| *TaGW2-6B* | Hap-I | 1.6 | 16.1 | 14.5 | 32.3 |
| | Hap-II | 9.7 | 17.7 | 40.3 | 67.7 |
| *TaGW2-6A* | Hap-6A-A | 11.3 | 24.2 | 45.2 | 80.6 |
| | Hap-6A-G | 0.0 | 9.7 | 9.7 | 19.4 |
| *NAM-A1* | *NAM-A1b* | 4.8 | 11.3 | 24.2 | 40.3 |
| | *NAM-A1d* | 6.5 | 22.6 | 30.6 | 59.7 |

The association of the alleles with phenotypes revealed that *Rht-1* haplotypes had a minor but significant effect on GFe and GZn contents (Figure 4). The *Rht-B1a/Rht-D1a* haplotypes had slightly higher GFe and GZn contents compared to haplotypes with any semi-dwarfing allele. Similarly, the effect of *Rht-1* haplotypes was much higher on PH and GY. The presence of *Rht-B1b/Rht-D1a* and *Rht-B1a/Rht-D1b* haplotypes reduced the PH from 103.3 to 96.1 and 90.7 cm, respectively. Contrastingly, these haplotypes significantly improved GY from 1.72 to 2.33 and 2.22 t/ha, respectively. The *TaSus2-2B* haplotype Hap-H had a significant and positive effect on TKW.

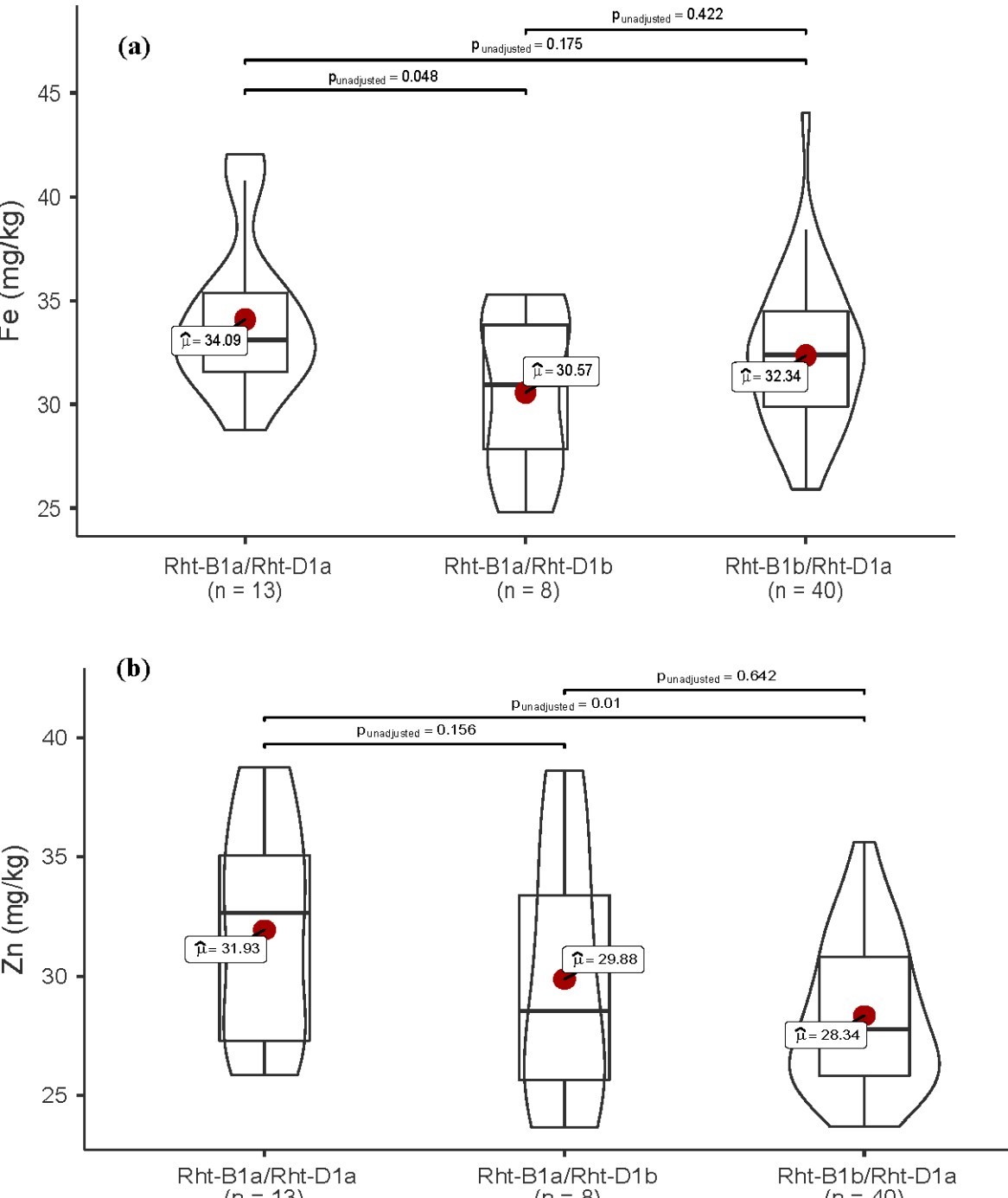

**Figure 4.** *Cont.*

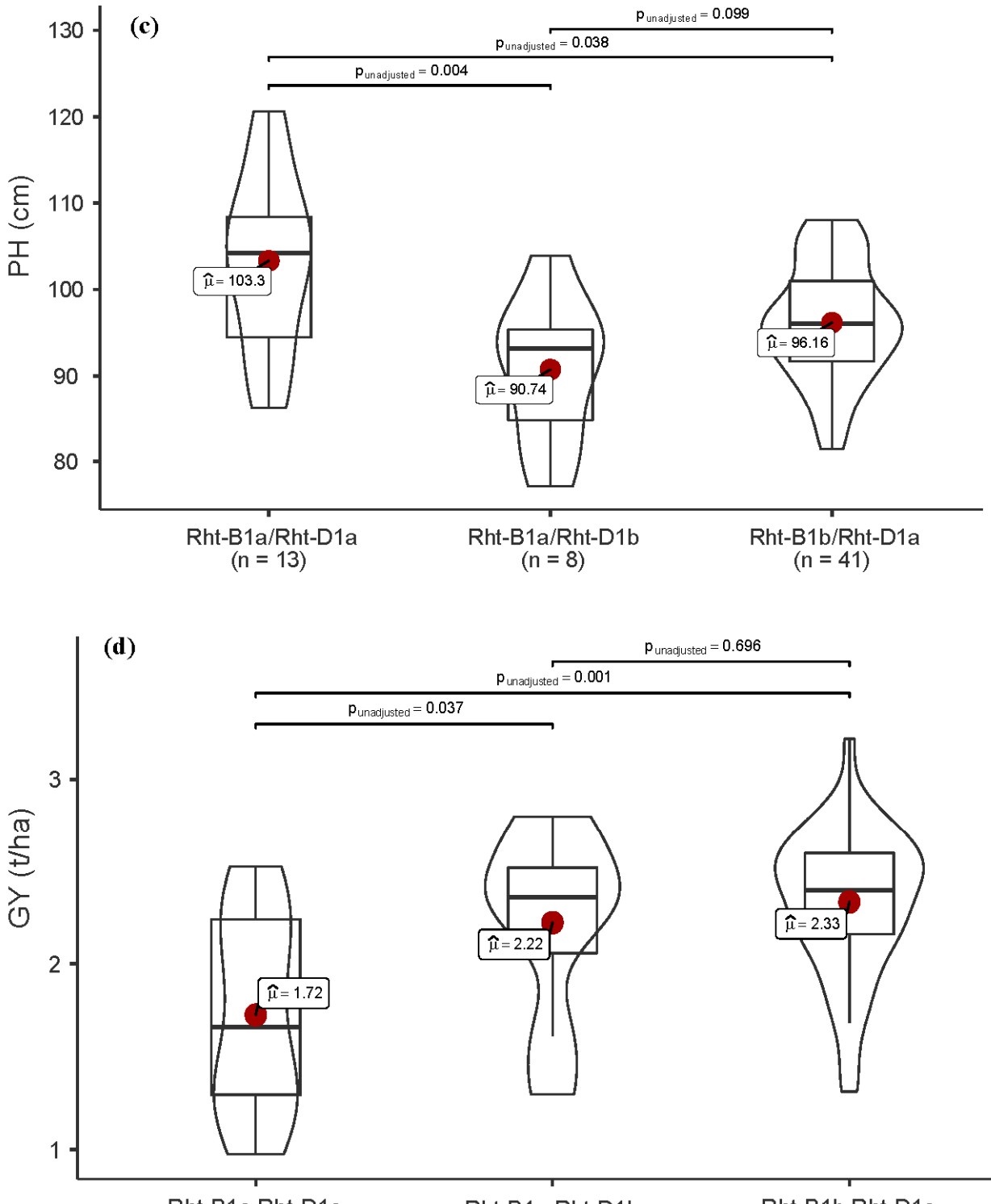

**Figure 4.** *Cont.*

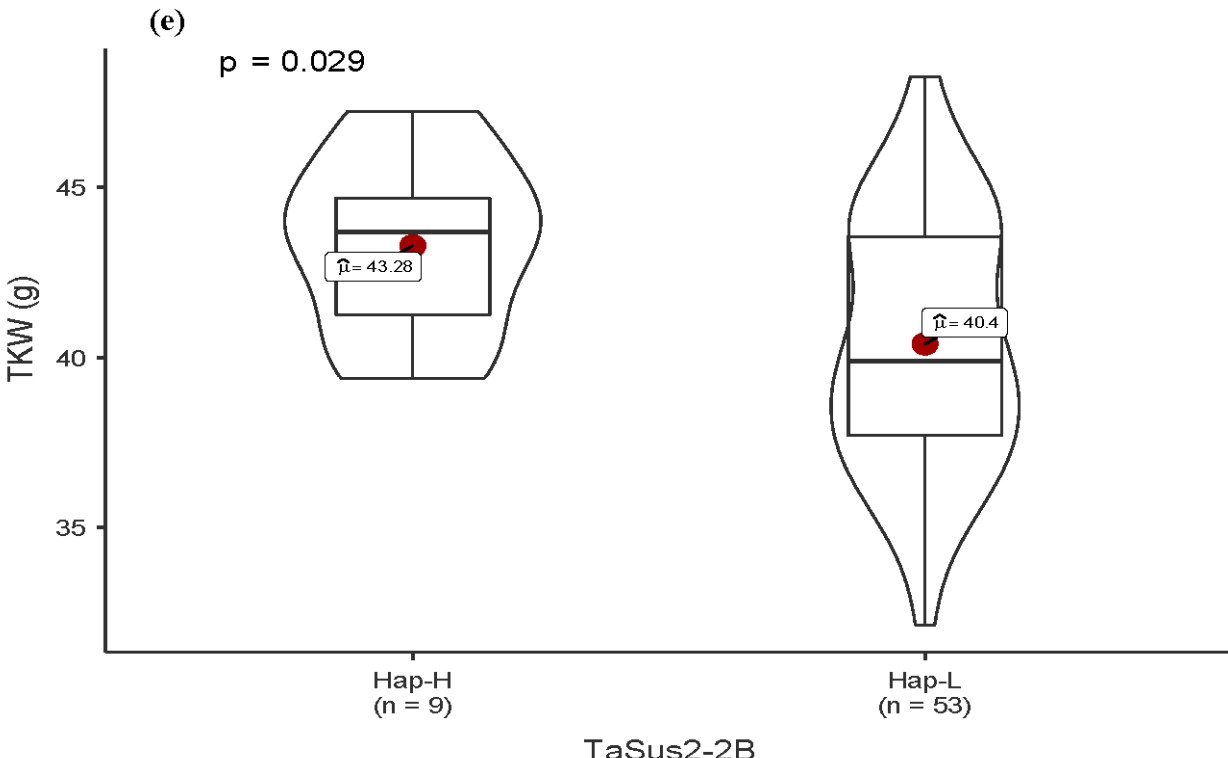

**Figure 4.** Allelic effects of *Rht-1* haplotypes on grain Fe (**a**), grain Zinc (**b**), plant height (**c**), grain yield (**d**) and *TaSus2-2B* gene on thousand kernel weight (TKW) (**e**) in historical wheat cultivars of Pakistan.

## 4. Discussion

A collection of historical wheat cultivars released over a period of 105 years were evaluated for yield-related traits and grain micronutrients. There were significant variations observed for all of the traits which indicated the progress in improving yield metrices over the course of selective breeding. The improvement in grain yield at the rate of 0.4% per year is relatively low compared to genetic gain in yield in other parts of the world [17,18]. The improvement in GY in Siberia over the period 1900 to 2010 was 0.59% [18], from 0.58 to 1.25%, in Great Plains hard winter wheat, while Gao et al. [16] reported a 57.5 kg ha$^{-1}$ yr$^{-1}$ gain in GY in Chinese bread wheat cultivars from 1950 to 2012 in the irrigated plain of China. The reason for the slightly slower rate of genetic gain is that the present comparison involved a relatively longer duration of 105 years, and the genetic improvement in yield in Pakistan was not temporally smooth. There were introductions of some high-yielding cultivars such as Pak-81 (1B.1R translocation) in the early 1980s, Inquilab-91 in early 1990s and their derivatives. These cultivars provided sudden increases in GY followed by stagnations in yield for many years. The current data support this hypothesis, showing that the CV (%) of the cultivars released after 2000 was half the CV (%) of the cultivars released in previous periods. This indicated the consistent progress made towards improving GY and the high stability of yield in modern cultivars.

Although significant progress has been made in improving productivity, there was limited progress in improving micronutrients in Pakistan and elsewhere. The cultivars released before the so-called Green Revolution in 1965 had relatively higher levels of GFe and GZn. It has been well established in previous studies that micronutrient concentrations decreased over time in modern wheat cultivars, and this has been validated in US hard winter wheat [19], historical and modern soft white wheat cultivars from US [9] and in a BroadBalk wheat experiment in the UK [7]. It was concluded that grain mineral concentration remained stable in wheat cultivars from 1845 to 1960s, while it significantly decreased in cultivars afterwards. However, some reports contradict this trend, as observed in the Siberian wheat cultivars where no significant change in mineral concentrations was

observed over a period of 110 years [18]. Our results are in complete agreement with these studies, and partially supported by the allelic effect of *Rht-1* haplotypes on GFe and GZn. Previously, analysis of GFe and GZn in several bread wheat and durum wheat near-isogenic lines of *Rht-1* genes revealed that semi-dwarf lines reduced GFe by 3.2 ppm and GZn by 3.9 ppm [8]. In another set of near-isogenic lines of *Rht-1* genes, semi-dwarf lines showed decreased levels of GFe and GZn, while K and Ca were increased [20]. However, no confounding effect of *Rht, Ppd* and *Vrn* genes was found to affect GZn and GFe concentrations in the association mapping panel of HarvestPlus [21], which supports our results that such effects could not be significant in natural germplasm.

In this study, the correlation between GFe and GZn was positive, which corroborates with most of the previous findings [22–24]. However, the extent of correlation is highly variable in most of the studies. The positive correlation between GFe and GZn is very useful to identify the common genetic basis to breed for both traits. It was important to observe that grain P was highly positively correlated with most of the micronutrients, including GZn, Ca, Mg, Mn and Se. In wheat grain, P is stored as phytic acid in aleurone, and significantly inhibits the bioavailability of divalent mineral cations [25]. Therefore, it is very important to devise biofortification breeding strategies to modify the distribution of P between phytic acid and inorganic P [24]. The principal component analysis suggested that most of the variation was explained by the first two principal components, and PC1 weighted towards the micronutrients with the highest loadings. Therefore, wheat cultivars with high scores for PC1 are likely to have high mineral concentrations, and most of the old cultivars are included in this category. The correlation of GZn with GY was not significant, but Cu and Al had significant negative correlations with GY. This contradicts most of the studies where a significant negative correlation was observed between GY and micronutrients [22,24,26]. However, the insignificant correlation of micronutrients with TKW has been reported elsewhere [22,27], which corroborates our results. It is important to carefully assemble the diversity collection for such relationships, as in our case the cultivar collection was from irrigated and rainfed areas from South and Central Pakistan, which have different yield-related attributes, such as TPP and TKW. Historically, the wheat cultivars from South Pakistan have more TKW on an average compared to cultivars from other parts of the country.

The target to increase GZn concentration by 12 mg/kg, and similarly increase GFe specifically for Pakistan, is very challenging. The approaches needed to enhance the GZn, GFe and other micronutrients will include the introduction diversity from other genetic resources or wild species of Triticeae [28,29]. In most cases, the landraces, synthetic hexaploidy wheat and wild relatives of wheat were identified with higher levels of grain micronutrients [30–32]. The conventional breeding approaches have been successfully used to incorporate such diversity into elite germplasm. CIMMYT's biofortification breeding program has developed elite cultivars by targeting crosses between high-yielding germplasm and high-micronutrient germplasm, and selecting the desired traits in large population sizes [4]. This strategy has resulted in the development and release of several cultivars, such as 'Zinc Shakti (Chitra)' in India, WB-02, HPBW 01 (PBW 1 Zn), Zincol-2016 in Pakistan, and BARI-Gom 33 in Bangladesh, having 33–40% more GZn compared to Czech cultivars [33].

Apart from the GFe and GZn, other micronutrients such as Ca, Se and Mn are also important for human health, and breeding for such micronutrients has been largely ignored. Our data show significant variations in other micronutrients. Selenium is an essential micronutrient with antioxidant, anti-cancer and anti-viral effects. In this study, grain Se concentration was positively correlated with GFe, GZn and TKW, and a two-fold increase was observed in some cultivars. Previously, significant variation was observed for Se concentration in bread wheat and related species, and as with other minerals, Se variation was associated with spatial variation in soil Se [34]. Similarly, the decrease in Ca concentration in modern wheat cultivars might have adverse health consequences, and biofortification for grain Ca is largely ignored [35]. The correlation was positive between Ca

and GFe and GZn and other micronutrients, which suggested common breeding strategies could be devised for simultaneous improvement of these micronutrients in wheat.

## 5. Conclusions

Conclusively, this study provided insight into the mineral status and yield of wheat cultivars historically deployed in Pakistan. Overall, the improvement in GY was not translated into an improvement in micro- and macronutrients. Although GFe (0.06 mg/kg/year) and GZn (0.15% year) slightly declined in modern wheat cultivars compared to old cultivars, there are some high-yielding cultivars such as Zincol-2016 and AAS-2011 which have high levels of micronutrients. Elucidating the genetic basis of GY and micronutrient concentrations could help to develop cultivars with both improved yield and biofortification status.

**Supplementary Materials:** The following are available online at https://www.mdpi.com/article/10.3390/agronomy11061247/s1, Table S1: Genetic gain analysis for micronutrients and morphological traits in historical wheat cultivars of Pakistan, Table S2: BLUEs for all phenotypes and the genotypic data used in this study

**Author Contributions:** A.R., Z.H., X.X. and T.M. designed the experiment; M.S. (Muzzafar Shaukat), M.S. (Mengjing Sun), S.R. and M.A. performed the experiment; A.R. and Z.M. wrote the manuscript; S.N. and S.M. performed the genetic studies; Y.H., A.R. and Z.H. conceived the idea and edited the manuscript. All authors have read and agreed to the published version of the manuscript.

**Funding:** We acknowledge the National Natural Science Foundation for funding support under the Research Fund for International Young Scientists (31950410563).

**Institutional Review Board Statement:** Not applicable.

**Informed Consent Statement:** Not applicable.

**Data Availability Statement:** All the phenotypic and genotypic data are available as Supplementary Materials.

**Acknowledgments:** We acknowledge the National Natural Science Foundation for funding support under the Research Fund for International Young Scientists (31950410563).

**Conflicts of Interest:** The authors declare no conflict of interest.

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
