# Peer review of "Genetic Gain for Grain Micronutrients and Their Association with Phenology in Historical Wheat Cultivars Released between 1911 and 2016 in Pakistan"

_agronomy, doi:10.3390/agronomy11061247_

Round 1

Reviewer 1 Report

Dr Catherine REBUFA                                                                                      

Aix Marseille Univ, Avignon, CNRS, IRD, IMBE, UMR 7263

Équipe Biotechnologie Environnementale et Chimiométrie

Faculté des Sciences - Site de l'Etoile - Av. Escadrille Normandie Niémen - case 451

13391 Marseille cedex 20

Tel : +33(0)4 13 94 49 71 - E-mail : [email protected]

May 31, 2021

Ref: agronomy-1233941

Title: Genetic gain for grain micronutrients and their association with phenology in historical wheat cultivars of Pakistan released between 1909 and 2018 in Pakistan

Corresponding authors: Awais Rasheed and Zhonghu He

The authors proposed a detailed study of mineral status, yield and genotyping of a large collection of wheat cultivars deployed historically in Pakistan between 1911 and 2016. But the objective was not clearly defined, and different points need precisions.

Title

This is not 2018 but 2016 as stated in the body of the manuscript.

Abstract

Lines 22-25: What is the point of showing that the two analytical methods (ICP-OES and EDXRF) lead to different quantifications for Fe and Zn? The important thing is to insist on the difference in content obtained depending on the cultivar; it would be preferable to mention the coefficients of variation but to specify why these two analytical techniques were used here.

Lines 25-26: What is the correlation coefficient? between the two methods? I did not understand.

Line 30: it was written “ten micronutrients” but also 8 have been cited in bracket…

Introduction

Line 46: “HarvestPlus, 2021”: add a reference number and give the website link of this research program on agriculture for nutrition and health

Line 57: why Zn concentration of wheat grain must be > 50 mg? To cover the needs of an adult? a child?

Line 69: “grain Fe, ….and Mg decreased significantly”: in what proportions?

Line 78: “some cultivars with high grain Fe/Zn…”: in what proportions?

Line 88: why cite (or use) these gene markers particularly?

What is the final aim of this study? To help the most vulnerable populations by proposing them a local agriculture with specific wheat cultivars (a wheat breeding) allowing them to fight against mineral nutrient deficient. But it was impossible in the case of intensive agriculture because the yield predominates. This initiative needs to be supported by different organisms. Would this be a plausible approach in Pakistan?

Materials and Methods

Lines 94-96: “cultivars were evaluated for two year 2018-2019 (later as 2018) and 2019-2020 (later as 2019) in the field at National Agriculture Research Center (NARC), Islamabad, Pakistan 95 using a randomized complete block design (RCBD) with two replications”: chemical and genetic analyses have been realized on sowing of 2018 and 2019 years and an average was done? I don’t understand.

Line 97: why precise specifically soil electrical conductivity? Was it the only parameter monitored for soil quality? Were the seedlings sown under controlled agro-climatic conditions?

Table 1: Do the wheat cultivars proposed in Table 1 correspond to different varieties identified in Pakistan, of different geographical origin, imported varieties? Which varieties are still in use today? I don’t understand the expression “release year”: can you give additional information?

Lines 104-109: units were missing for certain measured parameters but appear in following tables. Here acronym of thousand grain weight is here “TGW” but further, “TKW” was always used.

Lines 118-124: what is the experimental error on the analysis of minerals with the two used methods?

Line 167: the reference “Gao et al. (2017)” must not appear here. A reference number is required.

Table 2:

  • complete the missing unit for “SL” for example. Modify the acronym “SpPS by SNPS;
  • chose TKW or TGW as written lines 104-115;
  • express results with the same precision and give the experimental error

Results

Lines 182-185: Why the 62 wheat cultivars were grouped into 3 categories before their statistical analysis. How does this temporal division relate to the history of cultivars? Why not have done a PCA on all the data without differentiating between time periods to find significant variables?

Box plots in figure 1 are unreadable.

Line 219. A value of R2 equal to 0.68 was cited while a value of 0.53 appears on figure 2. Same remark for Zn values.

Lines 237-238: No separation was really observed on figure 3a and this PCA showed that temporal groups chosen were not representative of wheat sampling. If separation that there is, what are the variables which separated pre-1965 cultivars as it was written?

Line 242: it is very difficult to observe on Figure 3b that C-273 cultivar was in admixture with some modern cultivars. Point this case on figure 3b because cultivar names are unreadable on X axis.

Line 250: In the title of figure 2, what does the expression “coefficient of determination”?

Line 292 : in table 5, precise that allelic frequencies were expressed in %.

Figure 4 is unreadable.

Discussion

Line 307: The time studied is just as long as that studied in the literature, so the “105 year” period is not a reason for the slightly slower rate for genetic gain as mentioned. Find another explanation.

Line 339- 344: the PCA correlation cercle was not communicated and then, discussion about variable correlations cannot be validated.

Lines 352-352: Are there any studies linking the wheat cultivar characteristics to the quality of Pakistan soil according to agro-climatic zones of this country

What conclusion can be drawn from the determination of Fe and Zn by the two analytical techniques used?

In conclusion,

Different experimental details may be corrected, and some figures may be revised for a better readability. The aim of this study needs to be better focused.

 In my opinion, this work can be published after revision.

Reviewer 2 Report

Aims   and   methods   are   clearly   described;   authors   represent   the   ideas   and   knowledge with sufficient theoretical background. The text is well written, the conclusions and choices made in the publications are easy to follow and understandable. Interesting topic, nicely finished work. The results are presented well.

Author Response

Thanks for approving the ms.